# Crop Establishment and Weed Control Options for Sustaining Dry Direct Seeded Rice Production in Eastern India

Sanjoy Saha [1,*], Sushmita Munda [1], Sudhanshu Singh [2], Virender Kumar [3], Hemant Kumar Jangde [1,4], Ashirbachan Mahapatra [1,4] and Bhagirath S. Chauhan [5]

1    ICAR-National Rice Research Institute, Cuttack 753006, Odisha, India; Sushmita.Munda@icar.gov.in (S.M.); hemant11march89@gmail.com (H.K.J.); ashirbachan@gmail.com (A.M.)
2    International Rice Research Institute, NASC Complex, New Delhi 110012, India; sud.singh@irri.org
3    International Rice Research Institute, Los Baños 4031, Philippines; virender.kumar@irri.org
4    Indira Gandhi Agricultural University, Raipur 492012, Chhattisgarh, India
5    Queensland Alliance for Agriculture and Food Innovation, The University of Queensland, Gatton, QLD 4350, Australia; b.chauhan@uq.edu.au
*    Correspondence: ssahacrri@gmail.com

**Abstract:** Dry direct seeded rice (DSR) has emerged as an economically viable alternative to puddled transplanted rice to address emerging constraints of labor and water scarcity and the rising cost of cultivation. However, wide adoption of DSR is seriously constrained by weed management trade-off. Therefore, the availability of effective weed control options is critical for the success and wide-scale adoption of DSR. A field study was conducted at ICAR-National Rice Research Institute, Cuttack, India, in the dry seasons of 2015 and 2016 to evaluate the performance of three crop establishment methods and five weed control practices on weed management, productivity, profitability and energetics of dry DSR. The results demonstrated that weed density and weed dry weight was lower in drill seeding than broadcast seeding by 26–36% and manual line-seeding by 16–24%, respectively, at 30 and 60 days after crop emergence (DAE). Among herbicides, post-emergence application (17 DAE) of azimsulfuron was most effective in controlling weeds compared to early post application of bispyribac-sodium and bensulfuron-methyl+pretilachlor. Weed competition in the weedy check treatment resulted 58% reduction in rice yield. Among establishment methods, drill-seeding was most profitable with US $ 685 ha$^{-1}$ higher net income than broadcast seeding primarily due to higher yield. Among weed control treatments, azimsulfuron was most profitable resulting in US $ 160 and 736 ha$^{-1}$ higher net income than weed free and weedy check, respectively. The specific energy was lowest for drill seeding among establishment method and azimsulfuron among weed control practices, suggesting lowest energy consumed in producing per unit of grain yield.

**Keywords:** establishment methods; weed control; productivity; profitability; energetics; dry direct seeded rice

## 1. Introduction

Rice is the staple food for over half of the world's population hence called as "Global Grain". India contributes about 20% of total global rice production [1], therefore, the stability of rice production in India would play a key role in the world's food security. The coastal plain zone of eastern India is the major rice growing belt of the country but the flood-prone lowlands of east coast plains are highly diverse, complex and fragile in nature [2]. During the wet season, the crop experiences several abiotic stresses including drought, submergence, waterlogging, and flash floods along with the additional problems of salinity (in certain pockets) and cyclonic disturbances [3]. Rice cultivation during dry season (summer rice) offers a great potential for boosting and stabilizing the yield in the region [4]. The conventional method of rice crop establishment during the dry season in the region is manually transplanting rice seedlings in the puddled soil known as puddled

transplanted rice (PTR) that requires a large amount of water, labour, and energy, which are gradually becoming scarce and more expensive, making PTR more costly and less profitable.

Dry direct-seeded rice (DSR) has shown promise under the scenario of labour and water scarcity and is considered as a potential alternative to PTR [5–7]. Based on the previous studies, DSR saved 20-33% irrigation water compared to PTR [5]. It reduces the total labour requirement by 11–66% compared to PTR, depending on the season, location, and type of DSR [5,8]. The increased availability of short duration rice varieties has further encouraged farmers to explore this new method of establishing dry season rice in the coastal plain zone of eastern India [4].

Despite these benefits, however, the economic benefit from DSR is not realized many times by the farmers due to poor crop establishment and severe infestation of weeds. Risk of higher weed infestation and consequently higher yield losses is one of the major constraints in the wider-scale adoption of DSR and in the realization of full yield potential [9,10]. Weeds in DSR are major problem because weeds emerge concurrently with rice seedlings and hence rice does not get a head start as the case in transplanted rice. Also, early flooding to suppress the early flushes of weeds cannot be used in DSR as rice is sensitive to flooding at germination and early establishment stage. Therefore, high expenditure on labour for weeding, if rely on hand weeding, may further dampen the scope of any profit occurrence [11]. In DSR, the competition by weeds is so severe that yield losses may sometimes shoot up to 90% [12,13] resulting in concurrent economic loss.

There are several herbicides that are standardised and recommended for DSR all over the world. However, in India, there are legal restrictions in pesticide use and many of the popular herbicides viz., glyphosate, paraquat, butachlor, 2,4-D, oxyfluorfen, quinalphos, pendimethalin are under restricted use or recommended for ban [14–16]. Over the years, farmers have used oxadiazone, oxadiagyl, pretilachor and pendimethalin as pre emergence application in rice with reported weed control efficiency of 55% [17], 65–85% [18], 58–82% [19,20] and 30.4% [21] respectively. There are two main problems associated with these pre emergence (PRE) herbicides. Firstly, these herbicides suppress the weeds only till three weeks, but, the subsequent flush of weeds cannot be controlled in dry DSR. Secondly, many of the area under rice cultivation are rainfed which are vulnerable to extremities of weather conditions. Sudden rains after sowing result in damage of emerging rice seedlings [22]. So, the farmers have only limited options of herbicide use to control weeds throughout the critical period. Additionally, sole use of herbicide do not guarantee complete weed control, the farmers need to modify the cultivation practices to achieve desirable weed control. Manipulation in establishment methods hold high potential in reducing the weeds pressure. Weed flora composition changes significantly with alternation in rice establishment methods [5]. Earlier, Bhurer et al. [23] reported variation in yield reduction under different establishment methods that varied by 40–76% in broad-cast seeding and 20% in drill seeding. Therefore, an integrated approach involving the manipulation of crop husbandry combined with direct weed control using herbicides is expected to address the issue of weed infestation in dry DSR.

In DSR, the weed competition and weed control cost depend greatly on how the crop is established and how weeds are managed [24]. In DSR, the rice can be established by line sowing (manually or by using a drill) or broadcasting which can have differential effects on weed occurrence, crop growth, and rice yield. Information on weed dynamics and weed management under different DSR establishment methods in the coastal region of eastern India is limited. Hand weeding 2–3 times during crop growing season has traditionally been the common practice for weed control in this region. However, recently, because of the rising scarcity of labor, particularly their non-availability at a critical time, hand weeding is either delayed or not done optimally. Moreover, labor wages are rising, making hand weeding uneconomical. Integration of herbicides offers an alternative option to achieve timely and cost effective weed control in DSR. There are limited studies, especially in relation to systematic comparison of weed infestation, weed control efficiency of herbicides

applied at early or late post-emergence, and rice yield in different DSR establishment methods. Therefore, a field study was conducted with the objective to evaluate the effect of DSR establishment methods and weed management practices on weed control, rice yields, energetics and economics in the coastal belt of Odisha, India.

## 2. Materials and Methods

### 2.1. Site Description

The field experiment was undertaken at the Research Farm of the ICAR-National Rice Research Institute, Cuttack (20.5 °N, 86 °E and 23.5 m above mean sea-level), India, during the two consecutive dry seasons of 2014 and 2015. The soil of the experimental field was *Aeric* (Endoaquept) with sandy clay loam in texture, slightly acidic to neutral with pH (using 1:2.5, soil: water suspension) 6.79, total carbon 0.71%, available nitrogen 209 kg ha$^{-1}$, available P 17.8 kg ha$^{-1}$, and available K 121 kg ha$^{-1}$. The soil test was based on samples taken from the upper 20 cm depth just prior to start of the experimentation.

### 2.2. Experimental Design

The experiment was laid out in a split-plot design with three replications. Three establishment methods viz., drill seeding using a seed drill, manual line-seeding with 15-cm row spacing and broadcast seeding, were assigned to the main plots and five weed control treatments were in sub plots. Weed control treatments included bispyribac-sodium as early post-emergence (POST) herbicide at 30 g a.i. ha$^{-1}$, azimsulfuron as late POST herbicide at 35 g a.i. ha$^{-1}$, and currently recommended early POST ready-mix herbicide bensulfuron-methyl plus pretilachlor at 70 + 700 g a.i. ha$^{-1}$, along with weedy and weed free checks. Azimsulfuron is a broad-spectrum sulfonylurea herbicide recommended to suppress major grasses along with broadleaved and sedges. Bispyribac-sodium is a most widely used pyrimidinyl thiobenzoate herbicide in Indian subcontinent to suppress key grasses (for example, *Echinochloa* species and *Ischaemum rugosum*), broadleaf and sedges but not effective on grasses such as *Leptochloa chinensis*. Bensulfuron-methyl plus pretilachlor is the recommended herbicide mixture for broad spectrum weed control in both wet and dry DSR.

In an earlier study, late POST herbicide suppressed weeds effectively in dry DSR and azimsulfuron applied at 15 DAE showed very good efficacy (91% weed control efficiency) against complex weed flora particularly late emerging grass weed, *Leptochloa chinensis* [25]. Moreover, in recent years, *L. chinensis* has become a major weed in the vegetative stage of the rice crop. To compare the efficacy of early and late POST herbicides in the present study, azimsulfuron was applied at the 3–4 leaf stage of weeds (17 DAE), bispyribac-sodium was sprayed at the 2–3 leaf stage of weeds (10 DAE) and the ready-mix herbicide bensulfuron-methyl plus pretilachlor was applied at 3 DAE. In the weed free plots, weeds were removed manually at 15, 30, 45, and 60 DAE to keep the treatment free from weed competition.

### 2.3. Crop Management and Herbicide Application Details

The field was prepared by ploughing thoroughly with a disc plough followed by harrowing with a rotavator to get a fine tilth for ensuring easy movement of the seed drill on dry soil. The experimental field was divided into three replications each of them consisting three main plots (each having size 35 m × 25 m). Each main plot was divided into five sub plots (each having size 7 m × 5 m). The sub plots had gross plot size was 7 m × 5 m and the net plot size used for harvesting was 6 m × 4 m. The rice variety 'Naveen' (115 days duration, Indica type) was sown using a seed rate of 40 kg ha$^{-1}$ on January 14 and 15 during 2015 and 2016, respectively. For drill seeding, seeds were sown using a 9 row seed-cum-fertilizer drill developed at the ICAR- National Rice Research Institute (formerly CRRI). For manual line-seeding, seeds were placed in continuous furrows followed by a light harrowing to cover the seeds. A Furrow opener was used to make furrows. For the broadcast seeding treatment, seeds were broadcasted on the well pulverized soils followed by a light harrowing. First, a light irrigation was given immediately after seeding and the

field was kept saturated during the first 10 days. Thereafter, a thin layer of standing water (1–2 cm) was maintained for the next 21 days after rice emergence. Afterwards, irrigation water was applied at a 2–3 cm depth after disappearance of water from the field till 15 days prior to maturity.

In India, grasses and sedges are predominant weeds in DSR [26,27], for which bispyribac-sodium is profusely used till date [26,28]. Therefore, bispyribac-sodium is taken as check to compare the efficacy of new herbicide i.e., azimsulfuron. The efficacy of herbicide mixture i.e., bensulfuron-methyl plus pretilachlor was compared with bispyribac-sodium as the herbicide mixture is expected to have least/delayed chance of developing herbicide resistance in weeds [29].

Bispyribac-sodium and azimsulfuron were applied by spraying at 10 DAE and 17 DAE respectively on saturated soil (after draining out of water) using a knapsack sprayer fitted with a flat fan nozzle at a spray volume of 300 L ha$^{-1}$ and spray pressure of 200 kPa. The field was irrigated again after 48 h of spraying. The ready-mix herbicide, bensulfuron-methyl plus pretilachlor (in granular form) was applied 3 DAE after mixing with fine sand at 12 kg ha$^{-1}$ in saturated soil conditions. Full dose of $P_2O_5$ (50 kg ha$^{-1}$) and 2/3rd of $K_2O$ (33 kg ha$^{-1}$) were applied before sowing at the time of final land preparation and N (100 kg ha$^{-1}$) was applied in three equal splits, at 15, 35, and 55 DAE. All the other recommended agronomic and plant protection measures were adopted to raise the crop. The rest 1/3rd of $K_2O$ was applied along with the third dose of N.

*2.4. Field Measurements*

Observations on weed species were recorded at 30 and 60 DAE. At each sampling date, weed density was recorded species wise by placing quadrates of size 0.5 m × 0.5 m at two random locations in each sub plot. Weeds were cut at the ground level, washed with tap water, and oven dried at 70 °C for 48 h, before weighing. The dominant weed species were determined based on the summed dominance ratio (SDR) values expressed as percentage, computed using the following equation [30].

$$\text{SDR of a weed species} = [\text{Relative density (RD)} + \text{Relative dry weight (RDW)}]/2$$

where,

$$\text{RD} = (\text{Density of a given species}/\text{Total density}) \times 100$$

$$\text{RDW} = (\text{Dry weight of a given species}/\text{Total dry weight}) \times 100$$

Weed control efficiency (%) at 30 and 60 DAE were computed using the formula given below:

$$WCE = \left[\frac{(x - y)}{x}\right] \times 100$$

where, $x$ = weed dry weight in weedy check and, $y$ = weed dry weight in treated plots

Weed index was computed by using the formula given below:

$$WI = \left[\frac{(a - b)}{a}\right] \times 100$$

where, $a$ = yield in weed free plot and, $b$ = yield under treatment for which weed index is to be calculated.

Grain yield of rice along with other yield components were recorded at harvest at the 14% seed moisture content. Sampling was done from an area of 1 m$^2$ in each plot to determine above ground total dry weight (total biomass) and yield components. Panicles m$^{-2}$ was counted manually. Filled grains of 10 randomly selected panicles were counted to determine the number of grains per panicle. Biomass (sum of straw dry weight and grain dry weight) was calculated using grain and total dry weight of each treatment.

*2.5. Economics*

All the costs incurred for different field operations (tillage, seeding, irrigation, application of fertilizers and chemicals, harvesting and post-harvest operations) along with input costs (seeds, fertilizers and chemicals) were computed and summed up to obtain the total variable cost of cultivation. Sale prices of grain and straw based on prevalent market prices were summed up in each treatment to calculate the total revenue received from the sale of produce as gross returns. Net returns for each treatment were calculated by deducting the variable cost of cultivation from gross returns. The ratio between gross returns to total variable cost of cultivation was taken as benefit-cost ratio (B:C ratio) i.e., return per US $ of investment.

*2.6. Energy*

The energy consumption was calculated by multiplying the amount of input consumption with its unit energy equivalent as in Ziaei et al. [31]. From energy input and output; the net energy, energy use efficiency, specific energy and energy productivity were computed by following formulae [32,33].

$$\text{Energy input} = \text{Ehl} + \text{Epr} + \text{Emt}$$

$$\text{Energy output} = \text{Emp} + \text{Ebp}$$

$$\text{Net energy} = \text{Energy output} - \text{Energy input}$$

$$\text{Energy use efficiency} = \text{Energy output}/\text{Energy input}$$

$$\text{Specific energy} = \text{Energy input}/\text{Yield of rice}$$

$$\text{Energy productivity} = \text{Yield of rice}/\text{Energy input}$$

$$\text{Unit of Energy input and output} = \text{MJ ha}^{-1}; \text{Unit of yield of rice} = \text{kg ha}^{-1}$$

where Ehl, Epr and Emt refer to energy from human labour, energy from power and energy from materials such as seed, fertilizer, chemicals and irrigation, respectively. Emp and Ebp refer to energy from main product and energy from by product, respectively.

*2.7. Statistical Analyses*

Treatment $\times$ year interactions were non-significant for almost all the parameters, therefore, data of both years were pooled for analysis and average is presented. Data were analysed using analysis of variance (SAS Software packages, SAS EG 4.3) and means of treatments were compared based on the least significant difference (LSD) test at $p \leq 0.05$. Weed density and dry weight data were subjected to square root transformation and the transformed values were used in analysis. Correlation of weed dry weight, panicle numbers m$^{-2}$, number of grains per panicle, grain yield, B:C ratio and energy productivity of rice were determined using SAS EG 4.3.

**3. Results**

*3.1. Weed Composition and Weed Species Dominance Pattern*

The weed flora in the experimental plots had a mixed population of grasses, sedges and broadleaved weeds (Table 1). Among grasses, the dominant weeds were *Echinochloa colona* (L.) Link, *Leptochloa chinensis* (L.) Nees and *Digitaria sanguinalis* (L.) Scop. *Cyperus difformis* was the only sedge weed species present in the experimental plot. Among broadleaved weeds, following species were present: *Sphenoclea zeylanica* Gaertn., *Eclipta prostrata* L., *Alternanthera philoxeroides* Griseb., *Phyllanthus niruri* L., and *Ammannia baccifera* L. The weed species dominance pattern was found similar at 30 and 60 DAE except for *Cyperus difformis* L. It showed more dominance during 2016 over *L. chinensis* and *D. sanguinalis*.

**Table 1.** Weed composition in weedy plots and their summed dominance ratio (SDR) $\pm$ SE (standard error) (DAE).

| | At 30 Days after Crop Emergence | | | | | | At 60 Days after Crop Emergence | | | | | | | | |
|---|---|---|---|---|---|---|---|---|---|---|---|---|---|---|---|
| | *Echinochloa colona* | *Leptochloa chinensis* | *Digitaria sanguinalis* | *Cyperus difformis* | *Sphenochlea zeylanica* | *Eclipta prostrata* | *Echinochloa colona* | *Leptochloa chinensis* | *Digitaria sanguinalis* | *Cyperus difformis* | *Alternanthera philoxeroides* | *Sphenochlea zeylanica* | *Eclipta prostrata* | *Phyllanthus niruri* | *Ammannia baccifera* |
| Drill seeding | 15 | 10 | 8 | 9 | 6 | 3 | 18 | 14 | 12 | 11 | 2 | 9 | 4 | 0 | 3 |
| Manual line-seeding | 18 | 12 | 9 | 10 | 4 | 3 | 20 | 16 | 13 | 14 | 4 | 10 | 6 | 3 | 5 |
| Broadcast seeding | 21 | 14 | 13 | 12 | 6 | 3 | 24 | 17 | 15 | 15 | 4 | 11 | 6 | 5 | 6 |
| SDR | 28.5 $\pm$ 3.3 | 19.8 $\pm$ 2.3 | 17.0 $\pm$ 2.4 | 17.4 $\pm$ 1.5 | 9.1 $\pm$ 1.1 | 5.8 $\pm$ 1.2 | 22.2 $\pm$ 3.4 | 17.2 $\pm$ 1.8 | 14.9 $\pm$ 1.5 | 15.0 $\pm$ 2.0 | 4.2 $\pm$ 1.1 | 11.5 $\pm$ 1.3 | 6.7 $\pm$ 1.2 | 2.7 $\pm$ 2.3 | 5.7 $\pm$ 1.8 |

Based on SDR, grasses dominated at 30 and 60 DAE relative to sedges and broadleaves weeds (Table 1). For example, at both 30 and 60 DAE, the grass weed *E. colona* recorded the highest SDR value in the range of 22–29 followed by other grass species such *L. chinensis* (SDR value 17–20) and *D. sanguinalis* (SDR value 15–17), whereas broadleaf weeds had lower SDR values (in the range of 3–12). *C. difformis* had an intermediate of dominance with the SDR value in the range of 14–17. Among broadleaf weeds, *S. zeylanica* and *E. prostrata* were more dominant than other species.

### 3.2. Weed Density, Dry Weight, and Weed Control Efficiency

#### 3.2.1. Weed Density

Rice establishment methods and weed control treatments significantly influenced the weed density (Table 2). Among the establishment methods, both at 30 DAE and 60 DAE, weed density was highest in broadcast seeding plots followed by manual line-seeding plots and was lowest in drill seeding plots. At 30 DAE, weed density was 16% and 36% lower in drill seeding than manual line-seeding and broadcast seeding, respectively. At 60 DAE, a similar trend was observed with 14% and 26% lower weed density in drill seeding plots relative to manual line-seeding and broadcast seeding methods, respectively. Among weed control treatments, irrespective of rice establishment method, weed density at 30 and 60 DAE varied in the following order (averaged across years and stages): weedy (69 plants $m^{-2}$) > bensulfuron-methyl plus pretilachlor (42 plants $m^{-2}$) > bispyribac-sodium (35 plants $m^{-2}$) > azimsulfuron (24 plants $m^{-2}$). All the herbicides reduce weed density compared to weedy check ranging from 39% in bensulfuron-methyl plus pretilachlor, 49% in bispyribac-sodium to 65% in azimsulfuron-treated plots. There was no interaction effect of establishment methods and weed control treatments on total weed density at 30 DAE but the interaction was significant at 60 DAE suggesting herbicide effects varied with rice establishment method (Table 2; *p* value = 0.049). For example, under weedy treatment, weed density decrease in the following order: broadcast seeding > manual line-seeding > drill seeding. However, in herbicide-based treatments (bispyribac-sodium, azimsulfuron and bensulfuron plus pretilachlor), weed density did not differ between broadcast and manual line-seeding treatments but it was 28–31% lower in drill seeding plots relative to the broadcast seeding.

#### 3.2.2. Weed Dry Weight

Similar to weed density, weed dry weight was also influenced by rice establishment methods and weed control treatments (Table 3). At 30 DAE, weed dry weight in drill seeding and manual line-seeding plots was 24% and 16% lower than broadcast seeding, respectively. At 60 DAE, weed dry weight was 16% and 9% lower in drill seeding and manual line-seeding, respectively compared to broadcast seeding. Among weed control treatments, at both 30 and 60 DAE, weed dry weight was lowest in plots treated with azimsulfuron and maximum in weedy check and it decreased in the following order: weedy check > bensulfuron-methyl plus pretilachlor > bispyribac-sodium > azimsulfuron. Compared to weedy check, weed dry weight reduction was 63, 66 and 82% at 30 DAE in besulfuron plus pretilachlor, bispyribac-sodium, and azimsulfuron-treated plots, respectively. Similar pattern was observed at 60 DAE with 71, 75, and 85% in besulfuron plus pretilachlor, bispyribac-sodium, and azimsulfuron-treated plots, respectively. The interaction effect was significant among establishment method x weed control treatments (Table 3; *p* value = 0.029 at 30 DAE and 0.034 at 60 DAE). For example, in weedy plots, weed dry weight was highest in broadcast seeding, intermediate in manual line-seeding and lowest in the drill seeding plots at both 30 and 60 DAE. At 30 DAE, the effect of azimsulfuron was consistent among the establishment methods, but the effects of bispyribac-sodium and bensulfuron plus pretilachlor varied. However, at 60 DAE, in herbicide-based treatments, weed biomass did not differ by rice establishment methods within each herbicide treatment.

**Table 2.** Effect of establishment methods and weed control treatments on weed density (plants m$^{-2}$) at 30 and 60 days after emergence (DAE) at Cuttack, Odisha (Average of 2015 and 2016) [§].

| | Establishment Method (T) | | | | | | | |
|---|---|---|---|---|---|---|---|---|
| | **30 DAE** | | | | **60 DAE** | | | |
| **Weed Control Treatments (W) *** | **Drill Seeding** | **Manual Seeding** | **Broadcast Seeding** | **Mean *** | **Drill Seeding** | **Manual Seeding** | **Broadcast Seeding** | **Mean *** |
| | Weed density (plants m$^{-2}$) | | | | | | | |
| BPS | 20 | 26 | 35 | **27 [c]** | 34 | 43 | 49 | **42 [c]** |
| AZM | 12 | 16 | 21 | **16 [d]** | 26 | 32 | 36 | **31 [d]** |
| BSM + Pretl. | 26 | 32 | 43 | **34 [b]** | 41 | 52 | 57 | **50 [b]** |
| Weed free [†] | - | - | - | **-** | - | - | - | **-** |
| Weedy | 49 | 56 | 68 | **57 [a]** | 71 | 77 | 91 | **80 [a]** |
| Mean ** | **27 [C]** | **32 [B]** | **42 [A]** | | **43 [C]** | **51 [B]** | **58 [A]** | |
| | Analysis of variance (ANOVA) | | | | | | | |
| | *p* value | | LSD | | *p* value | | LSD | |
| Main plot (T) | <0.0015 | | 3 | | 0.0003 | | 3.0 | |
| Sub plot (W) | <0.0001 | | 3.6 | | <0.0001 | | 4.6 | |
| T × W | NS | | 3.4 | | 0.0494 | | 9.2 | |

BPS—Bispyribac-sodium (30 g a.i. ha$^{-1}$); AZM—Azimsulfuron (35 g a.i. ha$^{-1}$); BSM + Pretl.—Bensulfuron-methyl + Pretilachlor (70 + 700 g a.i. ha$^{-1}$); NS: not significant difference; [†] Weed free—Weed density was not recorded since weed was removed manually at 15, 30, 45 and 60 DAE; [§] Means are separated by least significant difference (LSD).* Within each timing, means with the same lower case letter in a column are not significantly different using LSD$_{0.05}$.; ** Within each timing, means with same upper case letter in a row are not significantly different using LSD$_{0.05}$. Data in bold are mean values of main plot and sub plot treatments.

**Table 3.** Effect of establishment methods and weed control treatments on weed dry weight (g m$^{-2}$) at 30 and 60 days after emergence (DAE) Cuttack, Odisha (Average of 2015 and 2016) [§].

| | Establishment Method (T) | | | | | | | |
| --- | --- | --- | --- | --- | --- | --- | --- | --- |
| | 30 DAE | | | | 60 DAE | | | |
| Weed Control Treatments (W) * | Drill Seeding | Manual Seeding | Broadcast Seeding | Mean * | Drill Seeding | Manual Seeding | Broadcast Seeding | Mean * |
| | Weed dry matter (g m$^{-2}$) | | | | | | | |
| BPS | 4.0 | 4.4 | 5.3 | 4.6 [c] | 19.2 | 20.9 | 23.4 | 21.1 [c] |
| AZM | 2.0 | 2.5 | 2.9 | 2.5 [d] | 10.5 | 13.0 | 15.1 | 12.9 [d] |
| BSM + Pretl. | 4.4 | 4.9 | 6.1 | 5.1 [b] | 22.5 | 24.4 | 26.6 | 24.5 [b] |
| Weed free [†] | - | - | - | - | - | - | - | - |
| Weedy | 12.4 | 13.4 | 15.4 | 13.7 [a] | 78.5 | 83.2 | 89.4 | 83.7 [a] |
| Mean ** | 5.7 [C] | 6.3 [B] | 7.5 [A] | | 32.5 [C] | 35.4 [B] | 38.7 [A] | |
| Analysis of variance (ANOVA) | | | | | | | | |
| | *p* value | | LSD | | *p* value | | LSD | |
| Main plot (T) | 0.0098 | | 0.70 | | 0.0013 | | 1.6 | |
| Sub plot (W) | <0.0001 | | 0.61 | | <0.0001 | | 2.5 | |
| T × W | 0.0286 | | 0.97 | | 0.0344 | | 4.6 | |

BPS—Bispyribac-sodium (30 g a.i. ha$^{-1}$); AZM—Azimsulfuron (35 g a.i. ha$^{-1}$); BSM + Pretl.—Bensulfuron-methyl + Pretilachlor (70 + 700 g a.i. ha$^{-1}$); [†] Weed free—Dry weight was not recorded since weed removed manually at 15, 30, 45 and 60 DAE; [§] Means are separated by least significant difference (LSD).; * Within each timing, means with the same lower case letter in a column are not significantly different using LSD$_{0.05}$.; ** Within each timing, means with same upper case letter in a row are not significantly different using LSD$_{0.05}$. Data in bold are mean values of main plot and sub plot treatments.

### 3.2.3. Weed Control Efficiency (WCE)

Weed control efficiency (WCE) of different weed management treatments in different rice establishment methods for 30 and 60 DAE are shown in Figure 1. The results showed that WCE was highest in azimsulfuron-treated plots irrespective of DSR establishment methods. These results indicated that late POST application of azimsulfuron at 17 DAE showed better performance (15–19% higher WCE at 30 DAE and 10–14% higher WCE at 60 DAE) than early POST application of bispyribac-sodium and bensulfuron plus pretilachor at 10 DAE and 5 DAE, respectively.

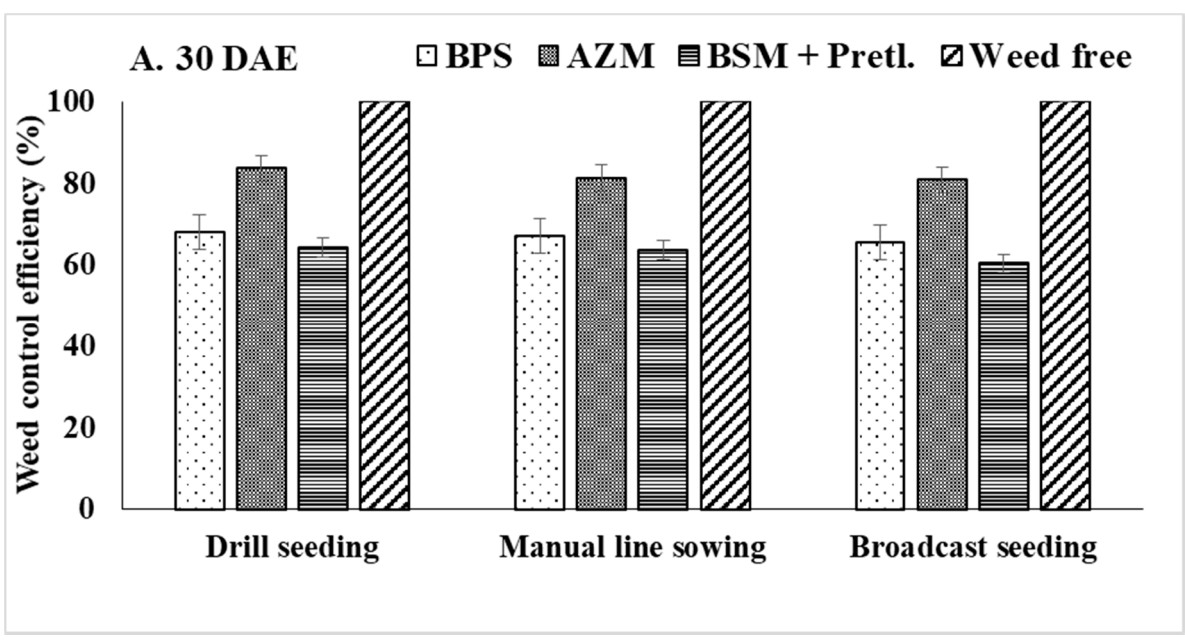

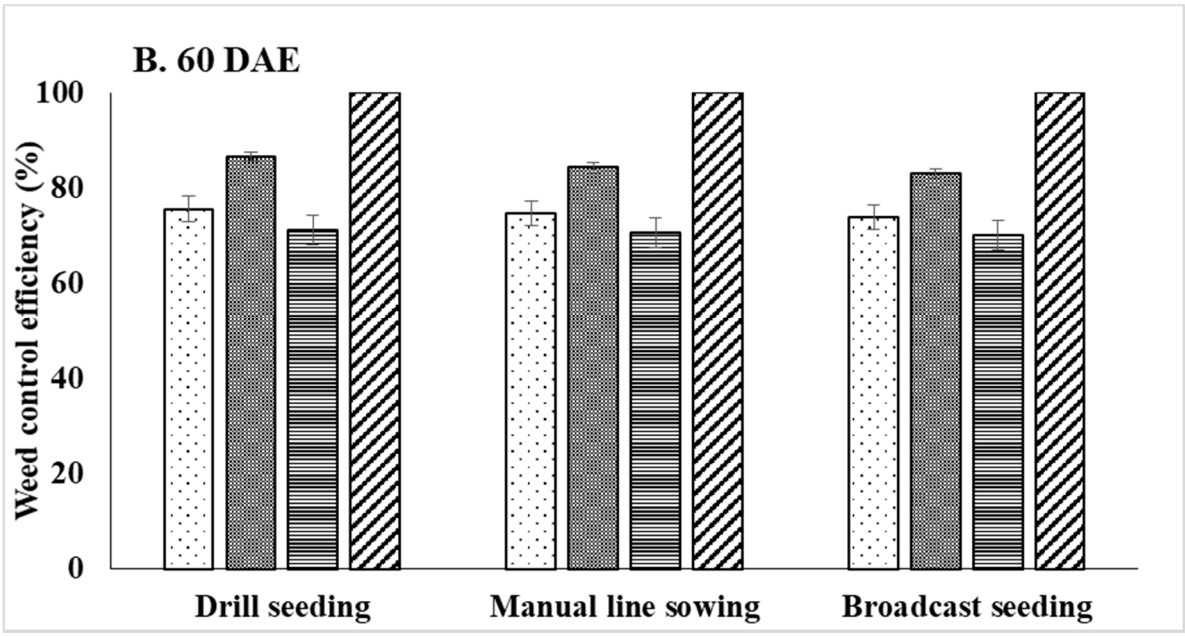

**Figure 1.** Effect of establishment methods and weed control treatments on weed control efficiency (%) at (**A**) 30 DAE and (**B**) 60 DAE at Cuttack, Odisha. Error bars represent standard error of mean. BPS—Bispyribac-sodium (30 g ha$^{-1}$); AZM—Azimsulfuron (35 g ha$^{-1}$); BSM + Pretl.—Bensulfuron-methyl + Pretilachlor (70 + 700 g ha$^{-1}$).

*3.3. Rice Grain Yield and Yield Components*

3.3.1. Rice Grain Yield

Rice grain yield was significantly influenced by both establishment methods and weed control treatments (Table 4). Based on two years average data, rice grain yield was highest in drill seeding (4.9 t ha$^{-1}$) followed by manual line-seeding (4.5 t ha$^{-1}$) and was lowest in broadcast seeding (3.9 t ha$^{-1}$). The weed control treatments also showed significant effects of different herbicide treatments on grain yield of rice. Among herbicide treatments, azimsulfuron treated plots recorded highest grain yield (5.2 t ha$^{-1}$) irrespective of rice establishment methods followed by bispyribac-sodium (4.9 t ha$^{-1}$) and was lowest in bensulfuron plus pretilachlor (4.4 t ha$^{-1}$). There was a drastic reduction in grain yield (58%) in weedy plots over weed-free check. Bensulfuron plus pretilachlor was found to be the least effective herbicide option in dry DSR with 15% yield reduction compared to azimsulfuron and 20% yield reduction compared to weed free plots. In azimsulfuron and bispyribac-sodium treated plots, yield reduction relative to weed free plots were only 5 and 11%, respectively.

A significant interaction effect of rice establishment methods and weed control treatments was recorded on grain yield (Table 4). In weed free and azimsulfuron-treated plots, yields of drill seeding and manual line-seeding was similar but was higher than broadcast seeding. However, in other treatments (bispyribac-sodium, bensulfuron plus pretilachlor, and weedy), yield varied with rice establishment methods in the following order: drill seeding > manual line-seeding > broadcast seeding.

3.3.2. Yield Components

The difference in rice grain yield was reflected on yield components. Yield components such as panicle number m$^{-2}$ and number of grains panicle$^{-1}$ were significantly influenced by rice establishment methods and weed control treatments (Table 4). However, the interaction of establishment methods x weed control treatments was not significant for these yield components. The highest number of panicles m$^{-2}$ was recorded in drill seeding (251) and it was reduced by 10% in the manual line-seeding and 17% in broadcast seeding as compared to drill seeding. Among the weed control treatments, the highest panicle numbers m$^{-2}$ was recorded in weed free plots (270) and it was similar to azimsulfuron-treated plots but higher than other treatments. Among, different herbicide treated plots, panicles m$^{-2}$ were similar in azimsulfuron and bispyribac-sodium treated plots but in besulfuron methyl plus pretilachlor treated plots; it was 14% lower than azimsulfuron. Herbicide treated and weed free plots had 24–56% higher panicles m$^{-2}$ than weedy check.

Grains panicle$^{-1}$ did not differ between drill seeding and manual line-seeding but were 18% higher in drill seeding than broadcast seeding (Table 4). Among weed control treatments, grains panicle$^{-1}$ were similar in weed free, bispyribac-sodium and azimsulfuron-treated plots but in bensulfuron-methyl plus pretilachlor treated plots, grains were lower than weed free and azimsulfuron-treated plots. Grains panicle$^{-1}$ were 32–38% lower in weedy plots compared to other weed control treatments.

**Table 4.** Effect of establishment methods and weed control treatments on yield components and grain yield of rice (Average of 2015 and 2016) [§].

| Weed Control Treatments (W) * | Method of Establishment (T) | | | | | | | | | | | |
|---|---|---|---|---|---|---|---|---|---|---|---|---|
| | Drill Seeding | Manual Seeding | Broadcast Seeding | Mean * | Drill Seeding | Manual Seeding | Broadcast Seeding | Mean * | Drill Seeding | Manual Seeding | Broadcast Seeding | Mean * |
| | Panicles m⁻² | | | | Grains Panicle⁻¹ | | | | Grain yield | | | |
| | m⁻² | | | | | | | | t ha⁻¹ | | | |
| BPS | 254 | 230 | 212 | 232 [bc] | 90 | 84 | 79 | 84 [abc] | 5.4 | 4.9 | 4.3 | 4.9 [c] |
| AZM | 274 | 249 | 230 | 251 [ab] | 95 | 87 | 82 | 88 [ab] | 5.7 | 5.3 | 4.6 | 5.2 [b] |
| BSM + Pretl. | 235 | 211 | 199 | 215 [c] | 87 | 82 | 77 | 82 [c] | 4.8 | 4.3 | 4.0 | 4.4 [d] |
| Weed free [†] | 294 | 270 | 247 | 270 [a] | 97 | 91 | 84 | 91 [a] | 5.9 | 5.5 | 4.9 | 5.5 [a] |
| Weedy | 197 | 170 | 153 | 173 [d] | 65 | 54 | 50 | 56 [d] | 2.8 | 2.2 | 1.9 | 2.3 [e] |
| Mean ** | 251 [A] | 226 [B] | 208 [B] | | 87 [A] | 80 [AB] | 74 [B] | | 4.9 [A] | 4.5 [B] | 3.9 [C] | |
| Analysis of variance (ANOVA) | | | | | | | | | | | | |
| | *p*-value | LSD | | | *p*-value | LSD | | | *p*-value | LSD | | |
| Main plot (T) | 0.0086 | 18 | | | 0.0332 | 8.0 | | | <0.0004 | 0.155 | | |
| Sub plot (W) | <0.0001 | 22 | | | <0.0001 | 7.0 | | | <0.0001 | 0.205 | | |
| T × W | NS | 38 | | | NS | 13.5 | | | 0.0152 | 0.405 | | |

BPS—Bispyribac-sodium (30 g a.i. ha⁻¹); AZM—Azimsulfuron (35 g a.i. ha⁻¹); BSM + Pretl.—Bensulfuron-methyl + Pretilachlor (70 + 700 g a.i. ha⁻¹); NS: not significant difference; [†] Weed free—Dry weight was not recorded since weed removed manually at 15, 30, 45 and 60 DAE; [§] Means are separated by least significant difference (LSD). The LSD value under interaction compares establishment method means at same weed management treatment. * Within each timing and year, means with the same lower case letter in a column are not significantly different using LSD$_{0.05}$; ** Within each timing and year, means with same upper case letter in a row are not significantly different using LSD$_{0.05.}$ Data in bold are mean values of main plot and sub plot treatments.

### 3.4. Weed Index

The magnitude of yield reduction (%) due to weed competition under different treatments in comparison with weed free check is represented by weed index (WI). Average over two years, the lowest WI was recorded in azimsulfuron-treated plots when crop was established by drill seeding (4%) (Figure 2). The WI varied as follows in different DSR establishment methods viz., broadcast seeding > manual line-seeding > drill-seeding. The WI in different weed management treatments varied as follows viz., weedy > bensulfuron-methyl plus pretilachlor > bispyribac-sodium > azimsulfuron.

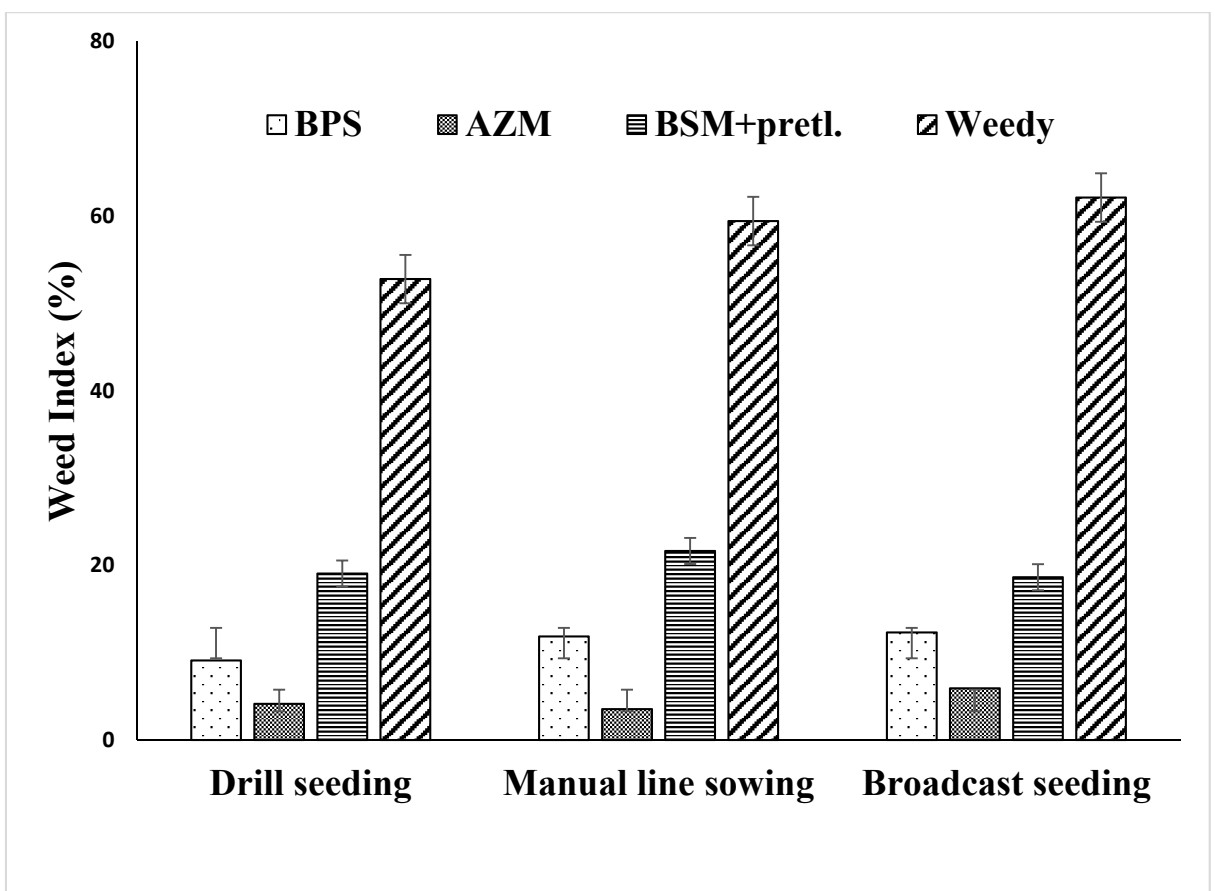

**Figure 2.** Effect of establishment methods and weed control treatments on weed index (%) at harvest. Error bars represent standard error of mean. BPS—Bispyribac-sodium (30 g ha$^{-1}$); AZM—Azimsulfuron (35 g ha$^{-1}$); BSM + Pretl.—Bensulfuron-methyl + Pretilachlor (70 + 700 g ha$^{-1}$).

### 3.5. Economics and Energy Balance

#### 3.5.1. Economic Analysis

Based on a two-year average, the results on economic analysis showed that the cost of cultivation was quite high in manual line-seeding (US $ 631) compared to drill seeding (US $ 599) and broadcast seeding (US $ 577) under different rice establishment methods (Table 5). Cost of cultivation did not differ in herbicide-based treatments but in weedy check, cost of cultivation was US $ 54 to 49 ha$^{-1}$ lower than herbicide-based treatments and US $ 249 ha$^{-1}$ lower than weed free check.

**Table 5.** Economics of dry direct seeded rice as influenced by different establishment methods and weed control treatments.

| Treatment | Cost of Cultivation | Gross Return | Net Return | B:C Ratio |
|---|---|---|---|---|
| | | ————————US$ * ha$^{-1}$———————— | | |
| | | Main plots [Method of establishment (T)] | | |
| Drill seeding | 599 | 1289 [a] | 685 [a] | 2.15 [a] |
| Manual line-seeding | 631 | 1171 [b] | 547 [b] | 1.86 [b] |
| Broadcast seeding | 577 | 1037 [c] | 460 [c] | 1.80 [b] |
| | | Sub plots [Weed control treatments (W)] | | |
| BPS | 571 | 1279 [b] | 709 [b] | 2.24 [b] |
| AZM | 571 | 1369 [a] | 801 [a] | 2.40 [a] |
| BSM + Pretl. | 576 | 1149 [c] | 586 [d] | 1.99 [c] |
| Weed free | 771 | 1437 [a] | 649 [c] | 1.86 [d] |
| Weedy | 522 | 594 [d] | 73 [e] | 1.14 [e] |
| | | LSD ($p \leq 0.05$) | | |
| Main plot (T) | | 61 | 34 | 0.11 |
| Sub plot (W) | | 76 | 38 | 0.12 |
| T × W | | NS | 64 | NS |

BPS—Bispyribac-sodium (30 g a.i. ha$^{-1}$); AZM—Azimsulfuron (35 g a.i. ha$^{-1}$); BSM + Pretl.—Bensulfuron-methyl + Pretilachlor (70 + 700 g a.i. ha$^{-1}$); NS: not significant difference; * US$ = 66.90 INR (Date: 15.12.2015); Data with the same superscripted lower case letters in a column are not significantly different using LSD$_{0.05}$.

Among different establishment methods, the significantly higher net and gross return were found in drill seeding than manual line-seeding and broadcast seeding (Table 5). Drill seeding resulted in US $ 138 and US $ 87 ha$^{-1}$ higher net return relative to manual line-seeding and broadcast seeding respectively. Among weed control treatments, net return was lowest in weedy check and was highest in azimsulfuron-treated plots and varied in the following order: azimsulfuron (US $ 801 ha$^{-1}$) > bispyribac-sodium (US $ 709 ha$^{-1}$) > weed free (US $ 649 ha$^{-1}$) > bensulfuron + pretilachlor (US $ 586 ha$^{-1}$) > weedy check (US $ 73 ha$^{-1}$).

3.5.2. Energy Balance

There was a difference in energy requirements for different rice establishment methods as well as different herbicides treatments. The energy input ranged from 10,292 MJ ha$^{-1}$ in broadcast seeding to 10,355 MJ ha$^{-1}$ in drill seeded plots and 10,169 MJ ha$^{-1}$ in weedy plots to 11,032 MJ ha$^{-1}$ in weed free checks (Table 6). Significantly higher energy output was recorded in drill seeding (137,306 MJ ha$^{-1}$) over manual line-seeding and broadcast seeding methods. Among the herbicide treated plots, the energy output was significantly higher in azimsulfuron-treated plots over bispyribac-sodium and bensulfuron-methyl plus pretilachlor treated plots. The energy balance i.e., net energy also showed the similar trend as energy output under different establishment methods and weed control treatments. The highest energy use efficiency (EUE) was recorded in drill-seeding (13%) and it was significantly reduced in other two establishment methods (11–12%). Similarly, azimsulfuron-treated plots showed significantly highest EUE (14%) over other weed control treatments (6–13%). The energy productivity was shown similar trend as EUE. Among different establishment methods, the specific energy was significantly higher in drill seeding than in manual line or broadcast seeding. Among weed control treatments, it was significantly higher in azimsulfuron, bispyribac-sodium, and weed free treatments compared to bensulfuron-methyl plus pretilachlor and weedy check treatments.

**Table 6.** Energy balance and productivity as influenced by establishment methods and weed control treatments.

| Treatment | Energy Input | Energy Output | Net Energy | Specific Energy | Energy Use Efficiency | Energy Productivity |
|---|---|---|---|---|---|---|
| | ————————————-MJ ha$^{-1}$———————————- | | | | ——%—— – | ——kg MJ$^{-1}$—— |
| | *Main plots [Method of establishment (T)]* | | | | | |
| Drill seeding | 10,355 | 137,306 | 126,951 | 2.31 | 13 | 0.47 |
| Manual seeding | 10,465 | 124,476 | 114,011 | 2.61 | 12 | 0.42 |
| Broadcast seeding | 10,292 | 110,289 | 99,996 | 2.95 | 11 | 0.38 |
| | *Sub plots [Weed control treatments (W)]* | | | | | |
| BPS | 10,205 | 136,323 | 126,118 | 2.12 | 13 | 0.48 |
| AZM | 10,178 | 145,316 | 135,139 | 1.97 | 14 | 0.51 |
| BSM + Pretl. | 10,269 | 122,038 | 111,769 | 2.37 | 12 | 0.43 |
| Weed free | 11,032 | 15,2935 | 141,903 | 2.04 | 14 | 0.49 |
| Weedy | 10,169 | 63,505 | 53,336 | 4.62 | 6 | 0.22 |
| | *LSD (p ≤ 0.05)* | | | | | |
| Main plot (T) | - | 6439 | 5941 | 0.18 | 0.6 | 0.02 |
| Sub plot (W) | - | 8066 | 7432 | 0.17 | 0.8 | 0.03 |
| T × W | - | NS | NS | 0.30 | NS | NS |

BPS—Bispyribac-sodium (30 g ha$^{-1}$); AZM—Azimsulfuron (35 g ha$^{-1}$); BSM + Pretl.—Bensulfuron-methyl + Pretilachlor (70 + 700 g ha$^{-1}$); NS: not significant difference.

## 4. Discussion

Heavy weed pressure, occurrence of several flushes of weeds and lack of available strategy for weed control are some of the critical factors for low adoption of DSR in India over the decades. This demands the development and deployment of proper weed control strategy for DSR. Availability of new POST herbicides with broad spectrum weed control ability during the critical period of crop-weed competition opened the opportunities for DSR in recent time. However, a crop establishment method plays an important role in weed emergence, its subsequent growth and choice of herbicides for its control as well as economics and energy requirements are also influenced by establishment methods and type of weed control options.

### 4.1. Effect on Weeds

The highest total weed density and weed dry weight was observed in broadcast seeding compared to manual line-seeding or drill seeding methods. The possible reasons of higher weed incidence in broadcast seeding relative to drill seeding may be because of uneven/non-uniform crop stand in broadcast seeding compared to line-seeding manually or by drill. Ichikawa [34] also found severe weed pressure at early stage in broadcast seeding due to uneven and poor crop establishment which resulted in higher crop-weed competition in comparison to spot seeding and row seeding. Uniform crop establishment resulted from the congenial micro environment of rhizosphere in the drill-seeded crop [22] and fast initial growth favoured rice crop to compete with weeds and helped in smothering the weed flora.

Among the tested herbicides, significantly higher weed density and dry weight in the bensulfuron-methyl plus pretilachlor treated plots might be due to poor control of grasses, particularly late emergent ones. The density of grasses was very high in this treatment plots at 60 DAE (data not shown). Bensulfuron-methyl and pretilachlor, are reported to be very effective in transplanted rice where mixed population of weeds occurred [35]. The higher efficacy of bensulfuron-methyl plus pretilachlor was found in an earlier study in wet DSR during the dry season [36]. Bensulfuron-methyl alone is recommended in rice for many annual and perennial broadleaved weeds and sedges [37]. Luo et al. [38] reported that carbon source like sodium lactate is responsible for rapid degradation of bensulfuron-methyl making it less effective on weeds at later stages. Another report showed rapid



degradation of bensulfuron-methyl due to repeated application owing to adaptation of soil bacteria which can utilize bensulfuron-methyl as a source of carbon and energy [33].

Regardless of rice establishment methods, azimsulfuron provided better weed suppression relative to other two tested herbicide combinations (Table 3). Suppression of grasses (weed control efficiency 98.5%) along with complete control of sedges and broad-leaved weeds by azimsulfuron was also reported by Saha et al. [4]. In this study, it was observed that azimsulfuron performance was not affected by rice establishment methods but performance of other herbicide treatments varied with establishment methods. This differential response could be attributed to differential performance of herbicides with different levels of weed density as reported in different dry DSR establishment methods. The result suggests that azimsulfuron was less affected by differential density in different establishment methods, whereas the performance of bispyribac-sodium and bensulfuron-methyl plus pretilachlor was reduced under higher density in the broadcast seeding. Mahajan and Chauhan [39] reported higher efficacy of azimsulfuron in row-seeded rice over other herbicides (pendimethalin, bispyribac-sodium and fenoxaprop-p-ethyl). Bispyribac-sodium, the most widely used herbicide for control of grasses in rice, was found less effective against late emergent *L. chinensis*. Many studies have reported poor control of *L. chinesis* by bispyribac-sodium [5,40]. Bispyribac-sodium has shown minimal translocation and a large amount is retained in the treated plant leaves [41], that indicates the residue left in the soil only gets absorbed by the roots of weed species only if they have extensive roots. It may be the reason for relatively less efficacy of bispyribac-sodium compared to azimsulfuron.

Higher efficacy of azimsulfuron compared to bispyribac-sodium and bensulfuron-methyl plus pretilachlor in different rice establishment methods ensured the effectiveness of azimsulfuron for suppressing weeds even at late vegetative stages in DSR. Gradual and persistent degradation of azimsulfuron in soil might have helped in suppressing the weeds for longer period of time. The slow degradation of azimsulfuron was aided by neutral pH (pH 6.8) of the experimental soil [42]. Pinna et al. [43], reported faster degradation of azimsulfuron in acid soils compared to neutral and slightly alkaline soil. Again, in unflooded soils, azimsulfuron was characterized as exhibiting moderate to high persistence [44] which is generally associated with higher residual effect of azimsulfuron on weed species. The persistent of bispyribac-sodium is low to moderate in un-puddled field (DSR) whereas it shows moderately higher persistent in flooded paddy soils [45]. This indicates the capability of BPS to control weeds for longer periods in transplanted rice than dry direct seeded rice. High weed control efficiency of azimsulfuron in DSR indicated that the efficacy of herbicide was further influenced by crop establishment techniques. Bispyribac-sodium, applied as early as 10 DAE when the crop was too small to cover the space between the plants which led to its rapid photo-transformation and photo-degradation enabling the weeds to emerge in the second flush. Suppression of grasses (weed control efficiency 98.5%) along with complete control of sedges and broad-leaved weeds by azimsulfuron was also reported by Saha et al. [4]. Suppression of late flushes of weeds leads to higher efficacy of azimsulfuron. The reverse is true for bensulfuron-methyl plus pretilachlor which was applied as early post-emergent herbicide and completely failed to control the weeds in dry DSR. The main reason could be poor control of late flushes of weeds after an early degradation of this herbicide [24,38].

Manual seeding combined with azimsulfuron was found effective, however, it needs to be used with caution. Azimsulfuron belongs to sulfonyl-urea class of herbicides (ALS inhibitor) which is associated with very high potential to develop herbicide resistance in weeds. Knezevic et al. [46] and Palou et al. [47] have reported development of herbicide resistance in as liltle as 3–4 years. Therefore, herbicide rotation is highly recommended. Use of azimsulfuron is not intended to replace the existing herbicides but simply adding other options for the farmers to choose from on need basis.

*4.2. Rice grain Yield, Economics and Energetics*

The higher rice grain yields in drill seeding and manual line-seeding compared to the broadcast seeding method was mainly because of higher rice panicle number m$^{-2}$ and to less extent due to higher grains panicle$^{-1}$. The lower yield in broadcast seeding could be attributed partly to poor and uneven crop stand and higher weed incidence indirectly because of an uneven crop stand. Similarly, grain yield in weed control treatments are linked with weed control efficiency with highest yield in the weed-free plots where there was no crop-weed competition followed by azimsulfuron where weeds were effectively controlled and yields were lower in weedy check and bensulfuron-methyl plus pretilachlor where weeds competed with the crop because of poor weed control in these treatments.

The lowest weed index (WI) in drill seeding indicated relatively less competition offered by weeds in drill seeding over manual-line and broadcast seeding. This might be due to better rice establishment at early vegetative stage when sown by the seed-cum-fertilizer drill. Since the system ensures placement of rice seeds and fertilizer that favoured young rice seedlings to establish in a better way in comparison to manual and broadcast seeding where basal fertilizer was applied by broadcasting during final land preparation. Higher WI in bispyribac-sodium and bensulfuron-methyl plus pretilachlor treated plots over azimsulfuron indicated comparatively poor efficacy of early POST herbicides in dry DSR that resulted in a higher percentage of yield reduction owing to the presence of weeds in the field during the critical period of competition.

The higher net return and B:C ratio in drill-seeded rice compared to the broadcast method despite a slightly higher cost of cultivation was attributed to higher grain yields than the broadcast method. The higher net income in the drill seeding method than manual line-seeding method was attributed to the combination of higher yields and lower cost of cultivation. Although the rice yield was significantly higher in weed-free checks over all the herbicide treatment plots but the net return was relatively less than azimsulfuron and bispyribac-sodium treated plots due to higher cost of production by engaging more labour for weed control in the weed-free treatment. The B:C ratio of weed-free checks was least in comparison to all the herbicide-treated plots. This indicates that weed control by herbicides was the most economical way to control weeds in DSR fields. There were negligible differences in the cost of cultivation of different herbicide treatments, but azimsulfuron-treated plots showed relatively higher net returns, which were 11% and 27% higher over bispyribac-sodium and bensulfuron-methyl plus pretilachlor treatment plots, respectively. This was mainly because of higher yield resulted from better weed control in azimsulfuron-treated plots. Thus, selection of a suitable herbicide is one of the most important criteria for weed control in dry DSR. This information could be very useful to farmers.

A similar trend was observed in terms of energy use efficiency and energy productivity. The amount of EUE was higher under drill seeding that indicated that sowing by using a seed-cum-fertilizer drill is the best way of establishing rice under dry direct seeding over manual line and broadcast seeding. The higher EUE in azimsulfuron treatments showed the best way of controlling weeds under dry direct seeding. The energy productivity obtained under different establishment methods indicated that per each unit of energy consumption in the fields, 0.47, 0.42 and 0.38 yield units was achieved in drill seeding, manual line-seeding and broadcast seeding, respectively. Thus drill seeding showed its advantage over other two establishment methods. The same was true for azimsulfuron-treated plots; it showed much higher units of yield achievement over other weed control treatments. Specific energy is the reversal of energy productivity hence its lower amounts showed that lesser energy was used for the production of each yield unit. So establishment of crop by drill seeding and weed control by azimsulfuron was superior for rice production under dry direct seeding due to both energy productivity and specific energy.

## 5. Conclusions

Selection of a proper crop establishment method and application of an appropriate herbicide have a remarkable influence on weed control and crop yield. Establishment of

the rice crop using drill seeding was effective in attaining higher rice grain yield, B:C ratio and energy use efficiency in dry DSR when weeds were kept under control by late POST application of azimsulfuron (35 g ha$^{-1}$) applied at 17 DAE. Manual seeding combined with azimsulfuron was found effective in achieving higher rice yield compared to the normal practice of broadcast seeding and weed control by bispyribac-sodium during the dry season in the coastal plain areas of eastern India. However, economic advantage and energy productivity were much higher under drill seeding. The information generated from this study will encourage the farmers to grow DSR and realize higher profitability in the coastal plain zone of India.

**Author Contributions:** Conceptualization, S.S. (Sanjoy Saha), S.S. (Sudhanshu Singh) and V.K.; methodology, S.S. (Sanjoy Saha) and S.M.; analysis, A.M. and H.K.J.; investigation, S.S. (Sanjoy Saha) and S.M.; writing—original draft preparation, S.M., A.M. and H.K.J.; writing—review and editing, V.K., B.S.C. and S.S. (Sudhanshu Singh); funding acquisition, S.S. (Sudhanshu Singh). All authors have read and agreed to the published version of the manuscript.

**Funding:** This research was funded by International Rice Research Institute, Los Baños, Philippines.

**Institutional Review Board Statement:** Not applicable.

**Informed Consent Statement:** Not applicable.

**Data Availability Statement:** The data presented in this study are available on request from the corresponding author. The data are not publicly available due to privacy.

**Acknowledgments:** We express our thanks to Director, ICAR-National Rice Research Institute, Cuttack, India and International Rice Research Institute, Philippines for all financial and technical support.

**Conflicts of Interest:** The authors declare no conflict of interest. The funders had no role in the design of the study; in the collection, analyses, or interpretation of data; in the writing of the manuscript, or in the decision to publish the results.

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
