# Peer review of "Crop Establishment and Weed Control Options for Sustaining Dry Direct Seeded Rice Production in Eastern India"

_agronomy, doi:10.3390/agronomy11020389_

Round 1
Reviewer 1 Report
The manuscript entitled "Crop establishment and weed control options for enhancing productivity, profitability, and energy use efficiency of dry direct seeded rice in the coastal plain zone of eastern India" aims to evaluate the performance of three crop establishment methods and five weed control practices on weed management, productivity, profitability and energetics of dry DSR.
I consider this study interesting for readers.
The Abstract is written well with clear justification and proper explanation.
Nonetheless, I have some comments. Please, find the specific suggestions below:
- Tables should adjust on one page with all detail.
- Lines 229, 230, 231 – rewrite the statement in the same explanation manner. Choose one category for the explanation, either “higher/lower,” to avoid the confusion of readers.
- Lines 272-277, divide the sentence into 2/3 short sentences. Long sentences make the statement unclear. Check the other lines.
- It would be better to merge the result and discussion part.
- Reconsider the title of the manuscript.
- Lines 96-98, “Weed control treatments included bispyribac-sodium as early post-emergence (POST) herbicide at 30 g a.i. ha-1, azimsulfuron as late POST herbicide at 35 g a.i. ha-1, and currently recommended early POST ready-mix herbicide bensulfuron-methyl plus pretilachlor at 70 + 700 g a.i. ha-1, along with weedy and weed-free checks.”
Why the amount of the chemical application is not constant for all considered herbicides? If the dose is recommended, then add references.
- Overall data, presented data is old during 2015-2016, which inconsiderate the data authenticity.
Author Response
The Dated: 15/02/2021
From
Sanjoy Saha
Principal Scientist
Crop Production Division
ICAR - National Rice Research Institute, Cuttack, India-753006
E-mail: ssahacrri@gmail.com
Phone no.- +91 82496 67558
To
The Editor-in-Chief
MDPI Agronomy
Subject: Re-submission of research article (Manuscript ID: agronomy-1093271) – Reg.
Dear Sir
The authors are thankful for considering the article for resubmission. We are enclosing herewith the revised manuscript entitled “Crop establishment and weed control options for enhancing productivity, profitability and energy use efficiency of dry direct seeded rice in the coastal plain zone of eastern India” for consideration of publication in your esteemed journal. The corresponding author of this manuscript is I and all the authors (Sanjoy Saha, Sushmita Munda, Sudhanshu Singh, Virender Kumar, Hemant Kumar Jangde, Ashirbachan Mahapatra and Bhagirath S. Chauhan) who have contributed to the research and mutually agreed that it should be submitted to MDPI Agronomy. Submitted manuscript is a Regular Article. The research project was conducted in collaboration with International Rice Research Institute, Los Baños, Philippines.
I am enclosing below herewith the list of modifications based on the comments of Reviewer 1 (Dated: 13.02.2021). The modifications are highlighted in yellow in the main text.
I hope the manuscript will now be considered for publication in your esteemed journal.
|
S. No. |
Comments |
Revision |
|
1. |
The manuscript entitled "Crop establishment ……….." aims to evaluate the performance of three crop establishment methods and five weed control practices on weed management, productivity, profitability and energetics of dry DSR. I consider this study interesting for readers. The Abstract is written well with clear justification and proper explanation. |
Thank you for the encouraging comments. |
|
2. |
Tables should adjust on one page with all detail |
As suggested each table has been adjusted in one page (Please see uploaded PDF file). |
|
3. |
Lines 229, 230, 231 – rewrite the statement in the same explanation manner. Choose one category for the explanation, either “higher/lower,” to avoid the confusion of readers. |
As suggested, the required modifications have been made. Line No. 231-233 |
|
4. |
Lines 272-277, divide the sentence into 2/3 short sentences. Long sentences make the statement unclear. Check the other lines. - |
As per your suggestion, the sentences have been separated into short ones. Line No. 271-282 |
|
5. |
It would be better to merge the result and discussion part. - |
The authors feel that a separate result section will make the presentation more vivid. As most of that data are presented in two way table, a separate result section would be more comprehensive for the readers. However, the authors are ready to reorganize the section if required.
|
|
6. |
Reconsider the title of the manuscript. |
As desired the title of the manuscript has been revised as "Crop establishment and weed control options for sustaining dry direct seeded rice production in eastern India".
|
|
7. |
Lines 96-98, “Weed control treatments included bispyribac-sodium as early post-emergence (POST) herbicide at 30 g a.i. ha-1 , azimsulfuron as late POST herbicide at 35 g a.i. ha-1 , and currently recommended early POST ready-mix herbicide bensulfuron-methyl plus pretilachlor at 70 + 700 g a.i. ha-1, along with weedy and weed-free checks.” Why the amount of the chemical application is not constant for all considered herbicides? If the dose is recommended, then add references |
Bispyribac-sodium as early post-emergence (POST) herbicide at 30 g a.i. ha-1 was recommended and adopted by farmers through Agriculture Department of the country like Krishi Vigyan Kendras (Reference No. 1).
Azimsulfuron as late POST herbicide at 35 g a.i. ha-1 and early POST ready-mix herbicide bensulfuron-methyl plus pretilachlor at 70 + 700 g a.i. ha-1 were recommended by the institute, ICAR – National Rice Research Institute (NRRI) (Reference No. 2).
|
|
8. |
Overall data, presented data is old during 2015-2016, which inconsiderate the data authenticity. |
In spite of many advantages, dry direct seeded rice is still not gaining popularity in our country due to severe weed competition and the inability of the farmers to manage the weeds by manual weeding. The experiment was designed to address this persistent issue in dry direct seeded rice in Eastern India. The paper intends to solve this problem and therefore, the results of the experiment are very relevant even today. The authors strongly believe that the outcome of our work will definitely help the rice farmers of the country.
I do agree that there has been delay on our part in organizing and analyzing the data. Moreover, our region was hit by severe cyclone “Fani” in 2019 which added to further delay. We prepared the final manuscript in 2020 and submitted as early as possible.
|
Yours sincerely
Sanjoy Saha
Principal Scientist
Crop Production Division
ICAR - National Rice Research Institute, Cuttack

Reviewer 2 Report
See attached file.

Author Response
We do really appreciate the reviewer and editor's valuable comments. Please see the attached file.
